# Strike one hundred to educate one: Measuring the efficacy of collective sanctions experimentally

**Philipp Chapkovski**  *

National Research University Higher School of Economics, Moscow, Russian Federation

* fchapkovskiy@hse.ru

## Abstract

In this paper, we test whether sanctions applied to an entire group on account of the free-riding of one of its members can promote group cooperation. To measure the efficiency of such collective sanctions, we conducted a lab experiment based on a standard public good game. The results show that, overall, collective sanctions are ineffective. Moreover, when subjects are able to punish their peers, the level of cooperation is lower in the regime of collective sanctions than under individual sanctions. Both outcomes can be explained by a general disapproval of the collective responsibility for an individual fault: in the post-experimental survey, an absolute majority evaluated such regimes as unfair. While collective sanctions are not an effective means for boosting group compliance, there are nevertheless two insights to be gained here. First, there are differences across genders: under collective sanctions, men's level of compliance is significantly higher than under individual sanctions, while the opposite is true for women. Second, there were intriguing differences in outcomes between the different regime types. Under collective sanctions, a person who is caught tends to comply in the future, at least in the short term. By contrast, under individual sanctions, an individual wrongdoer decreases his or her level of compliance in the next period.

## Introduction

Collective sanctions (CS) are imposed on an entire group for either a crime or misbehavior committed by a single group member. [1] defines CS as "the negative treatment inflicted by authorities or by an outgroup upon an entire social group, in reaction to an offense committed by one or some of its members". Sometimes the term "implicated punishment is used interchangeably, describing the situation when "once a wrongdoer is caught, all the group members are punished, no matter whether the group members are cooperators or defectors." [2]. The origin of collective sanctions is often traced back to pre-modern or primitive societies where this was a key concept of law [3], and it is easy to come to the erroneous conclusion that, in modern life, collective sanctions are largely limited to military bootcamps and prisons (cases mentioned by [4, 5] in their theoretical works on CS). In fact, many policymakers claim that

**Data Availability Statement:** Data are available at the OSF data repository (https://osf.io/fa5xv/).

**Funding:** This work was prepared within the framework and funded by the Basic Research

Program at the National Research University
Higher School of Economics (HSE).

**Competing interests:** The authors have declared
that no competing interests exist.

one of the most efficient methods for dealing with crime or norm violations is to make the entire group to which the perpetrator belongs responsible for the misconduct. Examples include corruption among university professors [6] or deviancy in public schools [7].

The belief that collective sanctions can succeed in curbing norm violations is common not only among policymakers, but among academics. For instance, in a review of solutions to collective action dilemmas, CS are listed as a tool to boost informal control in a group: "A common control technique is to punish the whole group for some act committed by one of its members. If the punishment is severe, as it often is, this technique may be horrendously effective" [8]. Some theoretical works have shown that implicated punishment is highly efficient in promoting cooperation in the evolutionary perspective [2]. As we show later, these claims have not been rigorously empirically tested.

Despite being ostensibly effective, collective sanctions are rarely implemented. There are several reasons for this. First, their usage goes against the entire logic of modern justice, which is based on the idea of retribution. Retributive justice rejects the rational cost-benefit analysis that provides a basis for collective sanctions owing to their presumed efficiency for the sake of individual responsibility. Second, when the entire group is sanctioned for the misdeed of one member, norm-obedient members are likewise punished, which may demotivate them from continued norm-compliance.

Advocates of collective sanctions usually justify them by two different lines of argumentation [9]. First, it is argued that the other group members are guilty of negligence. They had a chance to prevent the antisocial actions of a team member, but preferred to remain idle. Since idleness in correcting a team member's behavior is treated as antisocial action in and of itself, collective sanctions are intended to correct inaction and to increase the degree of peer control.

The second argument adopts a radical consequentialism, viz. a rational cost-benefit analysis. On this view, it does not matter that group members are not directly guilty of the antisocial behaviors of the specific member: collective punishment is nevertheless warranted on the grounds of efficiency. As [10, p. 348] stated in his overview of collective sanctions: "*Group members might be punished not because they are deemed collectively responsible for wrongdoing but simply because they are in an advantageous position to identify, monitor, and control responsible individuals, and can be motivated by the threat of sanctions to do so.*" This logic is built upon the idea of delegation of responsibility by an outside authority. If the entire group is punished for the misdeed of one member, then it becomes the individual members' task to detect and prevent antisocial behavior. The positive consequences of delegating the responsibility to detect and prevent crime to the nearest neighbors of a perpetrator outweigh harms incurred by punishing innocents.

In both cases, the conclusion is the same: the introduction of collective sanctions transforms the task of an outside authority to find a wrongdoer (i.e., a free-rider in public good settings) into the responsibility of his/her peers to detect, prevent, and punish the perpetrator. This paper examines what kind of consequences collective sanctions have on cooperation within a group, and on the willingness of peers to punish uncooperative behavior. Thus, the main objective of this paper is to answer the following question: **can collective sanctions for an individual's antisocial behavior be beneficial for the norm of cooperation**?

The paper is organized as follows. In the next section, I list theoretical considerations for and against collective sanctions and how they may presumably affect norm-compliance, especially vis-à-vis the norm of cooperation. Then, I describe the design of the experiment, followed by its results. Finally, I describe some limitations of this experiment and compare it with results of similar studies.

## Theoretical arguments for collective sanctions

Incentives, both positive (i.e., rewards) and negative (i.e., punishments), have long been studied as an effective tool to promote cooperation. Meta-analysis [11] has shown that the cost and the source of incentives are two main factors that affect their efficacy. A distinction is made between decentralized incentives provided by peers holding a similar position within a group, and centralized incentives, imposed by an external authority. Decentralized punishment has been shown to be more effective than centralized punishment [11], whereas there was no difference in the efficacy of positive incentives between centralized and decentralized regimes. However, this depends on the legitimacy of the central authority: if it is elected by group members, sanctions are more effective than in decentralized cases [12]. Peer sanctions are prone to fall victim to anti-social punishment [13] (when defectors punish co-operators) but some mechanisms, such as a system of prior commitments, can curb this negative effect, making peer sanctions effective again [14]. This paper focuses on sanctions centrally imposed on the entire group in their interactions with decentralized peer-based individual sanctions.

Collective sanctions are described by some scholars as "a conventional legal tool that is efficient in many of its applications" [15, p. 453]. To-date, this and related claims have not yet been thoroughly tested empirically: to the best of my knowledge, so far there have been very few lab experiments testing the effects of collective sanctions on cooperation. This intuition is confirmed by other authors. In an unpublished paper, [16] mentions that his study "appears to be the first lab experiment involving collective punishment", and in their paper on random sanctioning, Fatas et al. claim that, "As far as we know, no experimental analysis of random punishment in teams has ever been done" [17] (sanctioning of a random member may be interpreted as a collective sanction: see more on this topic in the 'Discussion' section). The design used in [16] is the only directly comparable design to that presented in the current study (some other relevant studies are covered in the 'Discussion' section). There, participants engaged in a standard public-good game, where players chose how much money to invest in a group project. One out of five group members was randomly assigned the role of "central authority" and was able to punish other group members collectively. In some treatments, the interest of the central authority was aligned with the interest of the group: his or her earnings would increase with the amount invested in the group project. In the other treatment, the interests of the group and that of the enforcer were opposed. Dickson found that collective sanctions had a subtle, short-lived positive effect on cooperation in the case of aligned interests, and a strictly negative effect in the case of opposed interests. Although unlike the design presented in this paper, in [16] the principal was a part of the group, the probability of detection was 100% and the cost of punishment was not fixed (and thus, controlled for) across different treatments.

Despite the relative scarcity of empirical evidence, some theorizing emphasizes that collective sanctions may be an efficient tool for deterring people from free-riding and non-cooperation. There are three types of argument for collective sanctions: functional, preferential, and informational [18].

First, the functional argument claims that the introduction of collective sanctions increases the efficiency and willingness of other group members to conduct in-group policing. Via collective sanctions, a centralized norm enforcer delegates the power to detect norm violators downwards to the members of the group, as well as the authority to deal with wrongdoers on their own initiative. In order to do so, the central authority has to create sufficient incentives for group members to monitor their peers and enforce norms [19]. This is done through imposing sanctions on the entire group or distributing group-wide rewards: "*group members*

*have incentives to urge one another to seek out external sources of rewards and to comply with external dictates to avoid triggering externally induced punishments*" [20, p. 367].

Second, collective sanctions may work because they change the preferences of a wrongdoer, who realizes that if his or her norm violation is detected, additional punishment will be brought upon the other members of the group. If an individual cares about harm imposed on third parties, the prospect of causing others to be punished may deter him or her from engaging in the devious action in the first place.

Third, the informational argument states that it is hard for an external authority to detect who is guilty of antisocial behaviors, but group members usually know much more about their neighbors and the cost to them of identifying the violator is relatively low. Thus, the argument goes, collective sanctions may increase the rate of detection by group members, while the punishment itself can still be carried out by an external group. In this way, collective sanctions address the information asymmetry that exists between in-group and out-group members.

An additional argument, not fully covered by the typology presented in [18], is rooted in social identity theory, which explains cooperation and norm compliance through the commitment of an individual to the group he feels he belongs to [21]. People tend to cooperate more with their own group members in the wide range of behavioral games [22], including Dictator's game [23], and Public good game [24, 25] and the costly punishment of group members for norm violation is itself a second-order public good. If a person strongly associates him- or herself with the group, that may increase the "black sheep effect": the tendency to punish one's own group members more severely than outsiders [26, 27]. Collective sanctions, by producing a common negative experience for the group, would increase group cohesion, resulting in a larger "black sheep effect", raising the chances that norm violators are punished.

## Hypotheses

Collective sanctions can affect an individual's decision-making regarding free-riding in the production of a public good in two different ways: directly and indirectly. CS change individual preferences directly by increasing the cost associated with norm violation: the knowledge that someone else from the group will be punished for free-riding increases the moral costs of such an action [18].

On the other hand, when a collective sanction harms a cooperative person, despite not actually free-riding, this can produce a de-motivating signal that reduces willingness to cooperate in the future. The punishment of a co-operator can be interpreted as an antisocial punishment (even if not intentionally so) [13] and there is ample evidence that this kind of punishment significantly diminishes cooperation, both when such punishment is intentional [28], and when generated by a 'noisy' environment, which impedes the punisher from correctly identifying a free-rider [29].

Since these two effects are countervailing, the overall direct effect is thus unknown and depends on the degree of group cohesion [30] and the probability of being punished for the actions of others, which, in turn, depends upon the size of the group [31]. Group size is a crucial factor affecting levels of cooperation both directly and indirectly through the efficacy of external or internal sanctions. There is no clear-cut answer in the existing literature on the effect of group size on cooperation level. There are studies showing a strong positive effect of group size [32, 33], almost no effect [34] (for very large groups), or a curvilinear effect where the level of cooperation grows to a certain point and then declines [35]. Apparently, the overall effect is context-specific, depending on the nature of the game (in single-shot interactions, there is a positive effect of group size in public-good games, but not in *N*-person Prisoner's dilemma [36]) as well as on game parameters such as marginal per capita return [34].

The indirect effect of collective sanctions is a result of delegation and increased in-group policing. This effect relies upon the capacity of group members to police and punish each other for norm violation. Thus, the efficacy of CS interacts with another institutional choice: whether peers are allowed to punish free-riders or not. The indirect effect of peer punishment adjusts the information disparity that an external authority has with regard to the perpetrator, and makes people more inclined to deter their own group members from injurious behavior [15]. For instance, in credit markets with third-party liability like the Grameen bank program, each member of a group serves as a co-guarantor for everyone else in that group, which makes participants "*influence the other agents' costs of engaging in desirable and undesirable aspects*" [37, p. 155]

These two factors (direct and indirect) through which CS may affect the degree of norm compliance suggest the following hypotheses:

1. When peers are able to punish free-riders within their group, they will do it more frequently and to a greater extent under the threat of collective sanctions rather than when there is merely a threat of individual sanctions (IS);

2. In an institutional regime with peer punishment, due to the expected larger extent of peer punishment in CS the level of cooperation will be higher than in IS.

Under a regime of collective sanctions without peer punishment, there are two opposing processes described above: (1) the moral cost of free-riding increases, encouraging cooperation, and (2) co-operators are punished and thus de-motivated from further cooperation. Therefore, two alternative hypotheses need to be tested under collective sanctions (CS):

3a. Levels of cooperation under CS will be higher compared to those under IS, because of the higher moral costs of free-riding. 3b. The CS regime sends mixed signals to co-operators because, despite cooperating, they can nevertheless be punished. This mixed signal can reduce their willingness to cooperate in the future. Thus, under CS we should observe a lower level of cooperation than under IS.

## Method

The basic framework of this experiment was a standard public good game, which was played with and without peer sanctions, and with or without the possibility of collective sanctions. The game structure in general follows [2]: the stage of contributions in a public good game is followed by external monitoring with a certain probability, an implicated punishment mechanism is triggered if anyone in a group is found to be non-cooperative, and then peer punishment stage concludes. Unlike [2] in our design individuals make continuous rather than binary decisions regarding contributions, and peers could punish any other member of their group, not only defectors. The 2 × 2 design is represented in the Table 1, where the type of institutional regime (individual vs. collective sanctions for a failure to invest enough into a public good) is crossed with the presence or absence of ingroup policing.

The experiment consisted of 15 periods. Treatments with peer sanctions had three stages per period, and treatments without peer sanctions had two stages per period. Participants were divided into groups of three and were provided with an endowment of 20 tokens each. In

**Table 1. Treatments.**

|  | No peer punishment | Peer punishment |
|---|---|---|
| **Individual sanctions** | Baseline (IS; No peer) | IS; Peer |
| **Collective sanctions** | CS; No peer | CS; Peer |

Stage 1, they decided how much to invest into a group project. In Stage 2, an external check of individual contributions was performed. In Stage 3, participants could use deduction tokens for peer sanctions.

Group composition remained fixed across all 15 rounds (partner matching), but the identities of specific participants in a group were not revealed in order to avoid retaliative strategic punishment or non-cooperation across rounds. Participants were informed in advance of the game structure: number of periods, number of stages in each period, the size of the group and the permanence of participant pairings for the duration of the game. At the start, participants were also instructed about the exchange rate (10 tokens for 50 US cents). The specific instructions for each treatment are given in the 'Supplementary materials' section.

The first stage consisted of a standard public good game where individuals face the choice of whether to cooperate or free-ride. This part was the same for all four treatments, but the anticipation of possible consequences at later stages may influence a person's decision to contribute more or less at this stage, depending on the institutional regime (CS or IS) and potential peer sanctions at Stage 3.

Before participants initiated Stage 1, they were informed that if they should contribute less than a certain amount (specifically, 10 tokens or less), a possible check of contributions by a computer could negatively affect their payoff at Stage 2. Stage 2 is the only stage where CS and IS treatments differed. Each group's contributions were checked with the same probability (1/3, more details are provided below), but the consequences were different. In the case of individual sanctions, if a person did not meet a minimum threshold requirement, and the external check revealed this, he or she bore individual consequences. By contrast, in the case of collective sanctions payoffs were reduced for the entire group, if at least one individual did not meet investment requirements. The last, third stage appeared only in treatments with peer punishment.

To introduce an element of external authority that imposes collective or individual sanctions with a certain probability, we employed an automatic mechanism to periodically check whether individual contributions met a certain threshold.

This threshold was set at half of the total endowment: out of 20 tokens, 11 'should' be invested in the group project. This prescribed number was not presented to participants as a duty, and no morally loaded words (e.g. 'authority' or 'punishment') appeared in the instructions. Instead, participants were informed that their contributions would be checked with a certain probability. If contributions were found to be lower than a set threshold, their earnings for that particular period would be diminished (the exact text explaining this mechanism varied according to the specific experimental treatment).

The randomized checks were implemented as follows: A matrix of pre-generated random numbers from 1 to 100 was uploaded to the z-Tree server. Each group had an associated vector of 15 random numbers drawn from this matrix, one random number per period. In each period, if a number associated with this group and this period was less than 33, then the contributions of an entire group were checked. Thus, the probability that a given group's contributions were checked in each period was 1/3. For clarity and to avoid participant deception, this mechanism was explained to participants in a simplified manner; for example:

*In the second stage, there is a 33% chance that the contributions of everyone in your group are checked by a computer. Specifically, during every period, the computer generates a random number between 1 and 100 for each group. If the generated number equals or is lower than 33, then it checks the contributions of all group members in that group.*

Generating the numbers beforehand rather than during the experiment guaranteed that in each treatment there were the groups with a similar history of external controls. Since the order and frequency of external checks influences the decisions of individuals to cooperate and punish peers in subsequent periods, this design provided control for a history of 'checks' in each of our four treatments. The simplicity and clarity of automatically checking individual contributions came at the expense of some empirical authenticity: for the external authority specifically (whose role was taken here by the experimenter), the observation cost was zero. Since detecting and punishing violators came at no cost, the 'informational' factor of introducing collective sanctions mentioned above was missing. Nevertheless, we chose to implement this checking mechanism out of an overriding concern for simplicity.

The different sanction regimes were implemented as follows: In the individual sanctions (IS) regime, if the automatic computer check found an individual's contribution to be 10 tokens or less, **that participant's earnings** for that period were reduced by 7 tokens. If the group's contributions were not checked, then all individual earnings during that round were retained.

Under the collective sanctions (CS) regime, if the contribution of at least one group member was found to be 10 tokens or less during the automatic check, the earnings of **all group members** in that period were reduced by 7 tokens. Since a random number was assigned to the entire group to determine the computer checking, the mechanism was identical for both CS and IS regimes: either the entire group was checked for the amount of contributions they had made, or not. The only difference was in the sanctions, if contribution requirement was violated and was detected during the automatic check.

At Stage 3, in treatments with peer sanctions, participants were able to deduct points from other members of their group, up to a maximum of 10 points for each peer. Each deduction point reduced the recipient's earnings by 2 tokens, while also reducing the sender's earnings by 1 token.

Therefore the final payoff $\pi_i$ of an individual $i$ consists of three parts: a direct return from production of public good, peer punishment costs, and the cost of external sanctions.

A direct return from public good production $y - g_i + a \sum_{j=1}^{n} g_j$ was calculated as a difference between a fixed initial endowment $y$, individual investment in a public good $g_i$ and the total investment of all group members $\sum_{j=1}^{n} g_j$ multiplied by a rate of return $a$ which was 0.5 in our case. Peer punishment costs $k \sum_{j \neq i}^{n} (c p_j^i + p_i^j)$ were a sum of tokens individual $i$ spent to punish others $\sum_{j \neq i}^{n} p_i^j$ and sum of tokens other participants punished him with $\sum_{j \neq i}^{n} p_j^i$ multiplied by a punishing coefficient $c = 2$ and conditional ($k \in \{0, 1\}$) on presence of peer punishment in this treatment. The cost of external sanctions $F \cdot S(\vec{g_i})$, was calculated as an intensity of external sanctions $F$ multiplied by its probability $r$ and by a function $S(\vec{g}) \in \{0, 1\}$ that was equal 1 if the contribution of at least one member in CS (or just $i$-th member for IS) in the group did not meet the threshold.

$$\pi_i = y - g_i + a \sum_{j=1}^{n} g_j - k \sum_{j \neq i}^{n} (c p_j^i + p_i^j) - rF \cdot S(\vec{g_i}) \tag{1}$$

## Game-theoretical predictions

The expected amount of the fine imposed by a central authority (i.e. if a subject fails to invest above the necessary threshold of 10 tokens) is calculated as the probability of being caught ($p$), multiplied by the amount of the fine ($F$). A net loss of investment of the threshold $T$ is $(1 - a)$

$T$, where $a$ is the rate of return on investment to a common pool. Thus unless $pF > (1 - a)T$, a rational profit-maximizer will behave in the same way s/he would behave in a regime without a required minimum contribution. The same logic applies in the case when costly peer sanctions are introduced. These peer sanctions are a second-order public good, so there is an incentive to free-ride in their production. The purely game-theoretical (but certainly not behavioral) prediction is therefore that people do not make use of peer sanctions (as it is known from [13, 38] people *do* punish their peers ignoring these rational profit-maximizing considerations).

Thus, under an individual sanctions regime, on average the same equilibrium should be observed as in other standard public-good games with a peer punishment stage, no matter what preferences participants have towards the peer sanctions: if participants expect that non-cooperative behavior is punished by peers, then we should observe a convergence towards full cooperation, or, if people fail to provide this second-order public good, then cooperation will decline. When collective sanctions are applied, an optimal strategy depends on the size of j, an expected number of violators. Even if the probability of a group being checked is the same as it was under individual sanctions, the chances of being externally sanctioned grow with the expected number of wrongdoers. Above, we have already briefly described potentially complex relations between group size and levels of cooperation. Controlling for this cooperation-group size effect, the efficiency of collective sanctions may also vary with the growth of group size [31]: as the group gets larger, so do the chances of external sanctioning. In groups of a significant size under collective sanctions, norm compliance is not a viable strategy to avoid sanctions. Although as it was shown in [2], under some conditions, there is a curvilinear effect between group size and efficacy of collective sanctions where they are the most productive for the groups of intermediate size. The burden of being in such a group is increased because an individual participant is disadvantaged from an informational perspective: he or she may not know who was an actual perpetrator and so feels helpless, being punished by an external force without being able to identify the norm violator responsible for those sanctions. On the other hand, in smaller groups the introduction of collective sanctions increases the probability of peer punishment: thus, we can expect the growth of norm compliance. Since the vectors of these two mechanisms (lower cooperation rate in expectation of being punished even if you cooperate and higher expectation rate due to expected peer punishment) are opposed, without specific parameters (such as group size and expected frequency of norm violation) it is hard to give clear-cut theoretical predictions of whether the equilibrium would differ from an individual sanctions regime. This is relevant though only for one-shot public good games with collective sanctions: evolutionary both for finite and infinite populations introduction of collective sanctions theoretically results in growth of cooperation level [2].

The experiment was conducted in the Columbia Experimental Laboratory in the Social Sciences (CELSS) using the standard z-Tree [39]. The design was approved by the Columbia University Internal Review Board (IRB approval protocol number IRB-AAAQ5109), participants were recruited via the ORSEE online system. Before proceeding with the experiment, all participants signed a consent form according to the IRB protocol. Subjects were guaranteed that their decisions as well as their payoffs would remain completely anonymous. The number of participants in each of the four treatment groups is shown in Table 2. Instructions to participants and z-Tree code are given in Supplementary Materials.

## Results

Number of participants per each treatment is shown in the Table 2. The average payment the participants received at the end of the experiment was $22.00 (all currencies are US dollars),

**Table 2. Number of participants per treatment.**

| Treatment | Peer punishment | Collective sanctions | Participants | Observations |
|---|---|---|---|---|
| Baseline | No | No | 24 | 360 |
| IS; Peer | Yes | No | 30 | 450 |
| CS; No peer | No | Yes | 24 | 405 |
| CS; Peer | Yes | Yes | 27 | 405 |
| **Total** | | | **108** | **1620** |

including $5.00 as a reward for showing up. Earnings varied between treatments, being slightly higher for individual sanctions ($22.40 vs. $21.70 in CS) and for treatments without peer sanctions ($22.20 vs. $21.90), but statistically the difference was not significant.

As it can be seen from the Table 3 and Fig 1 average contributions pooled across all 15 rounds do not significantly differ between treatments with and without collective sanctions. The contributions in treatments with peer sanctions are substantially higher than without them. If we compare contributions in treatments with peer sanctions under two different regimes (IS and CS), CS results in slightly lower average contributions in contrast to what we would expect. Since contributions fail Shapiro-Wilk test for the normality of distributions, we tested the difference in averages using Kruskal-Wallis non-parametric test. It has detected no difference in average contribution levels between IS and CS for treatments without peer sanctions (p-value 0.43235) while showing that that under peer sanctions participants in "Collective sanctions" regime contributed significantly less (p-value 0.00977) than their counterparts in "Individual sanctions" regime.

The dynamics of individual contributions into a group project show similar patterns for the collective and individual sanctions regimes (Fig 2). All participants started with high contribution levels of 10 to 12 tokens out of 20. Without peer sanctions, cooperation began to decline steadily after the 5th or 6th round and by 15th round it reaches the level of 25% (5 or 6 tokens out of 20). With peer sanctions, the average contributions remained relatively stable at about half of the endowment (10-12 tokens) until the 15th (and the last) round, when the contributions dropped—a typical effect of the 'end game' for other Voluntary Contribution Mechanisms (VCM) with sanctions [40]. When peer sanctions were available, CS contributions were lower than IS. There was no such difference in CS and IS treatments without peer sanctions.

The subjects could choose to invest any number of tokens (between 0 and the total endowment of 20) into a group project, with the safe threshold of 11 tokens, below which an external punishment could be applied. In reality, their choice set was much more limited. 81% of contributions fell into one of three categories:

- 31% (502 observations) of contributions were 0, or total non-compliance;

- 28% (456 observations) of contributions were 11, exactly 'at the edge' of compliance;

- 22% (361 observations) of contributions were full cooperation of 20 tokens.

**Table 3. Mean contributions in PGG.**

| Treatment | Peer | CS | Mean | 95% confidence interval |
|---|---|---|---|---|
| Baseline | No | No | 8.20 | [7.54—8.86] |
| CS; No peer | No | Yes | 8.06 | [7.32—8.8] |
| CS; Peer | Yes | Yes | 10.03 | [9.3—10.76] |
| IS; Peer | Yes | No | 11.35 | [10.56—12.14] |

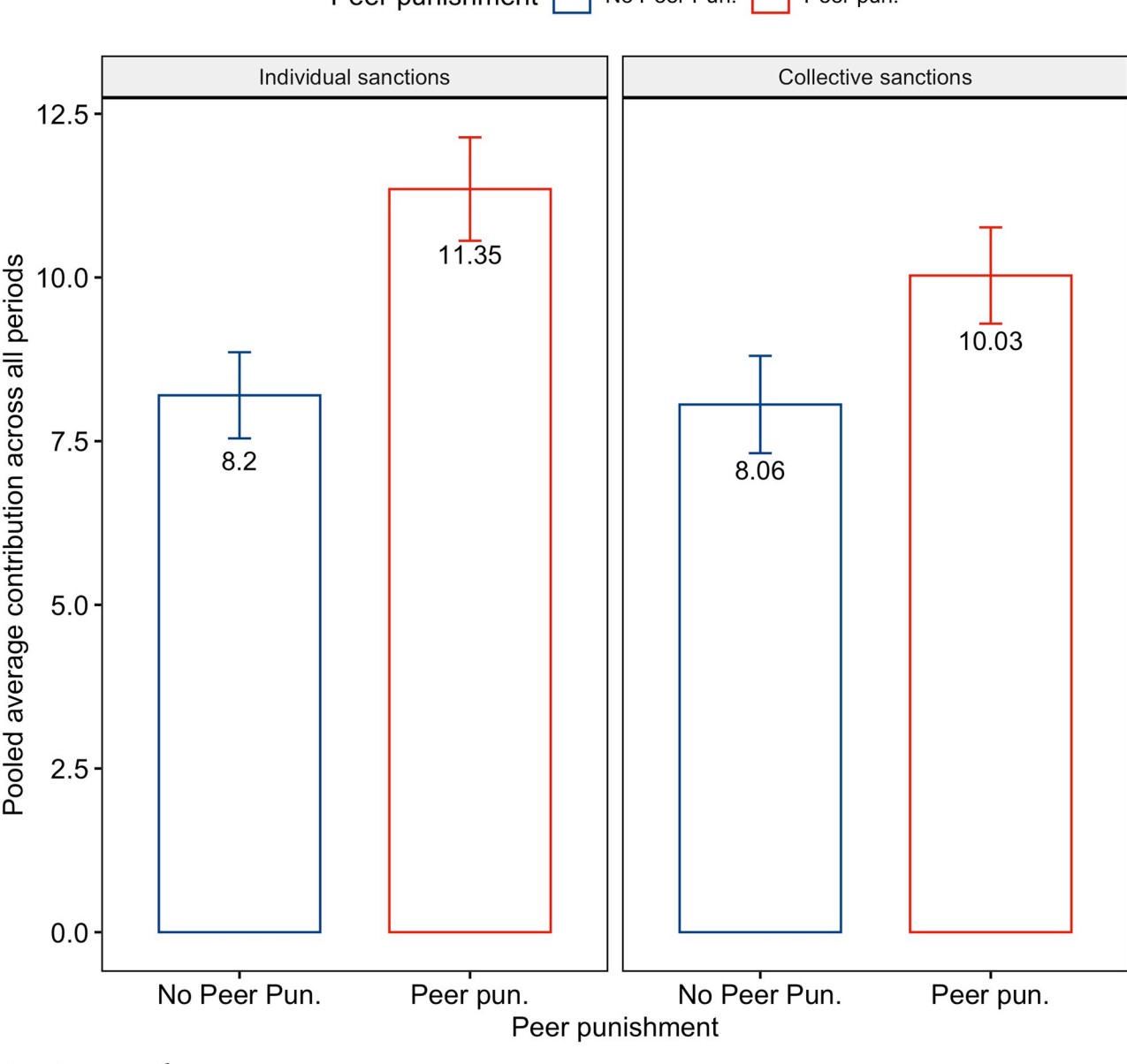

**Fig 1. Average contributions per treatment.**

This trimodal distribution (shown as a distribution of contributions in Fig 3) could provide an additional layer of analysis. When rules define a threshold for bare minimum cooperation, a rule-follower has a choice to be a marginal cooperator who contributes right above the necessary threshold, or to voluntarily cooperate to a degree larger than required. However contributions in public good games in general have the trimodal distribution where an overwhelming (93.8%) majority invests either 0, or all or exactly half of the endowment [41]. Thus the power of this analysis is pretty limited: we discuss these and other limitations in 'Discussion' section below.

The patterns of cooperation/non-compliance with regard to a threshold vary across the treatments (Fig 4). In general, CS again proves its ineffectiveness: the share of pure non-

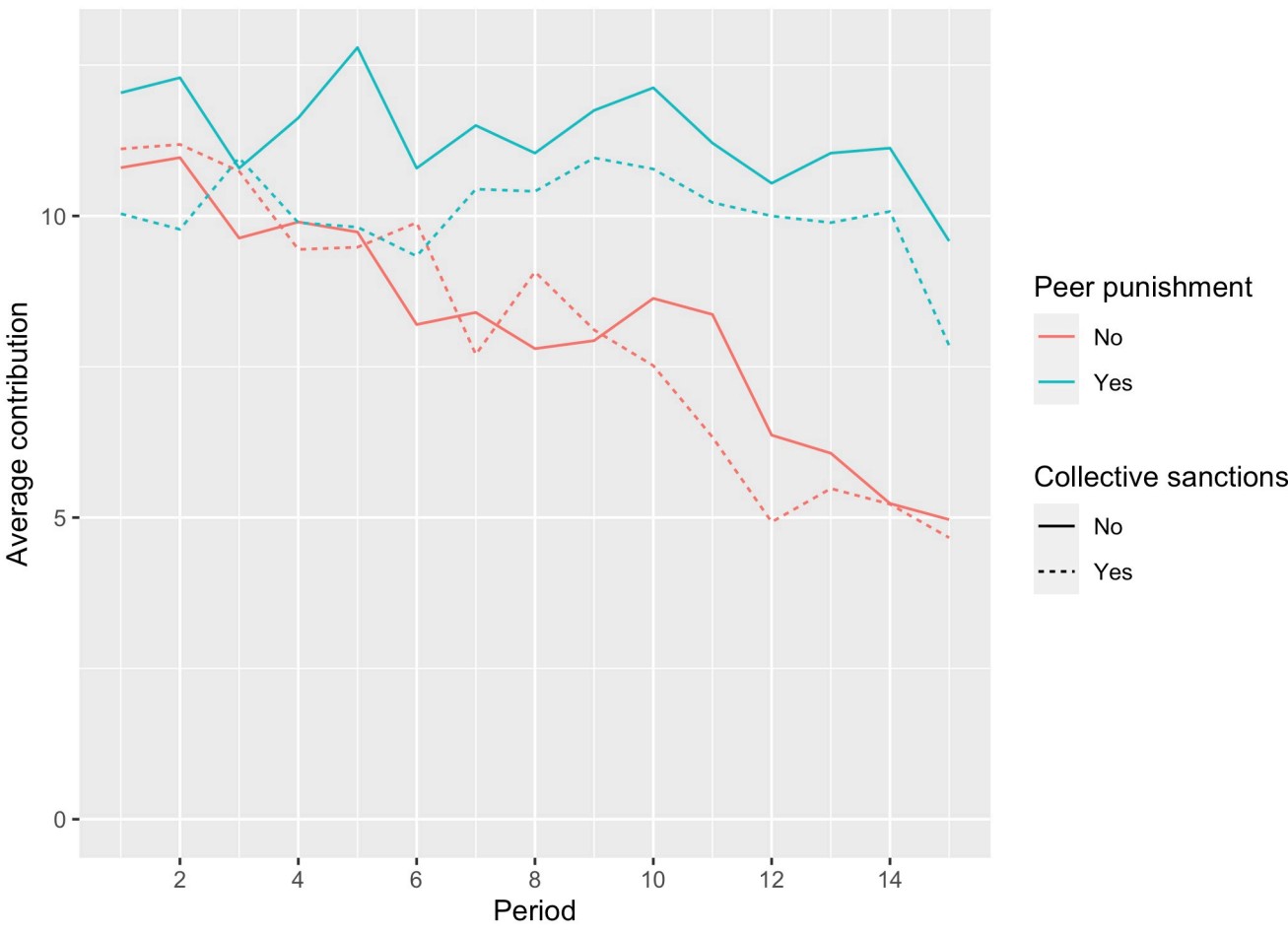

**Fig 2. Average contribution per period.**

compliers (those who contribute less than a threshold, $g_i < T$) is higher under collective sanctions than under individual sanctions. That is true for treatments both with and without peer sanctions. Without peer sanctions, the percentage of non-compliers under CS is 51% vs. 47% under IS, and with peer sanctions the share of non-compliers reach 36% under CS vs. 31% under IS.

There were clearly visible differences in behavior between genders across treatments with peer sanctions (bottom panel of Fig 5). Men contribute less than women in IS (on average 10 tokens vs. 12 for women) and significantly more in CS (15 vs. 8, or +75%)—see Table 4. No such pattern is observed in treatments without peer sanctions (top panel of Fig 5). The same is true if we look only at the contributions above the required threshold. On average, under IS, women who decided to 'obey the rules' invested 16 tokens, but invested only 12.8 tokens under a CS regime. The situation is exactly opposite for men (13.8 under IS vs. 17.0 under CS). The proportion of voluntary cooperators (investing strictly more than a required threshold) among women in IS is 64%, but only 27% among men. The situation is the opposite under collective sanctions, where 61% women are "bare" contributors, compared to only 24% of men.

In addition to the standard OLS models (for panel data with random-effects) we use random-effects Tobit-regression for panel data (Tables 5 and 6, respectively). We followed [42,

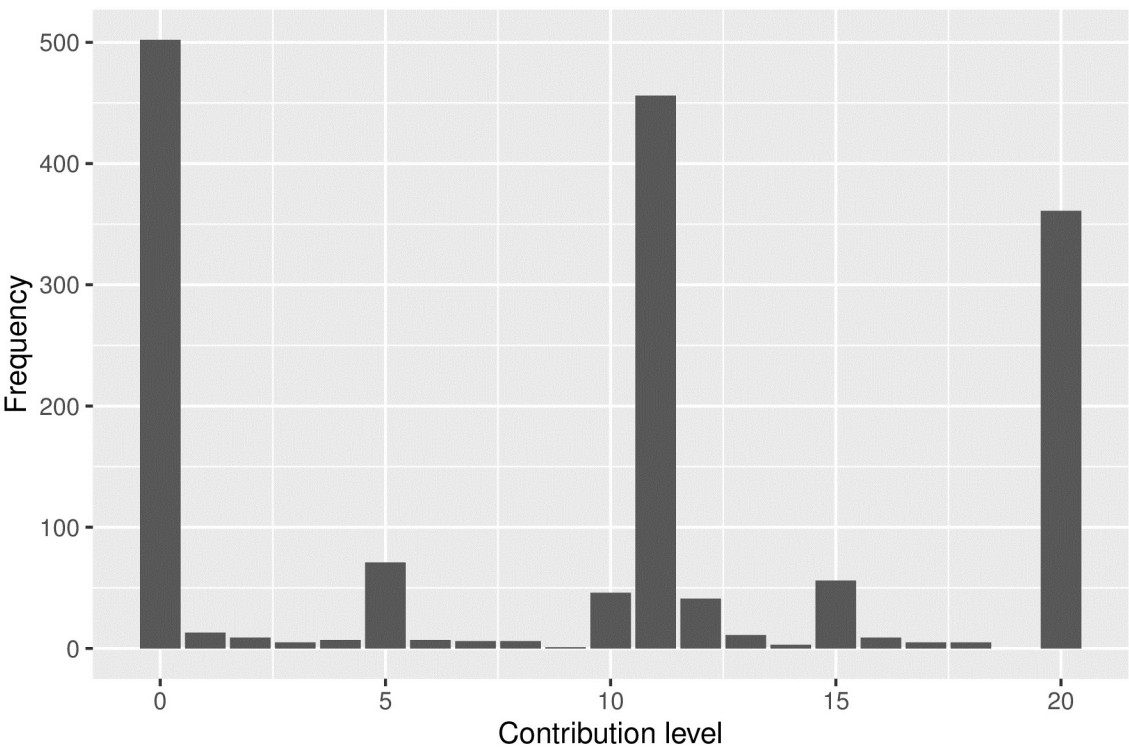

**Fig 3. Distribution of contributions.**

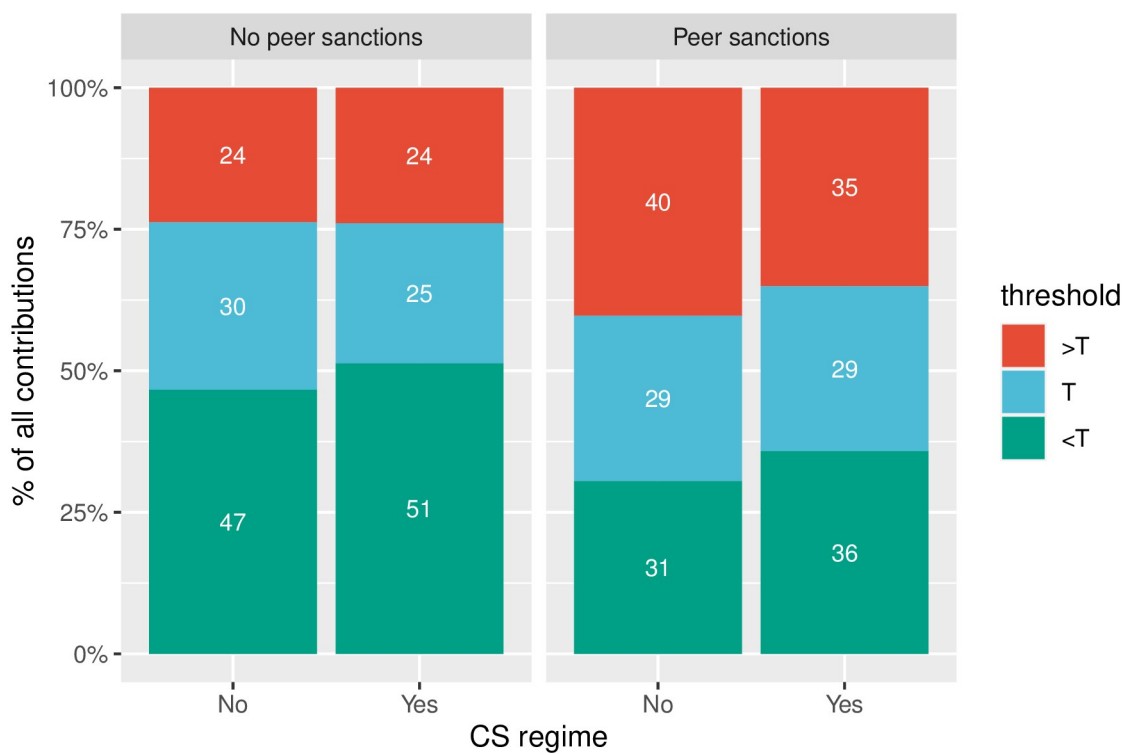

**Fig 4. Share of compliers and non-compliers across treatments.**

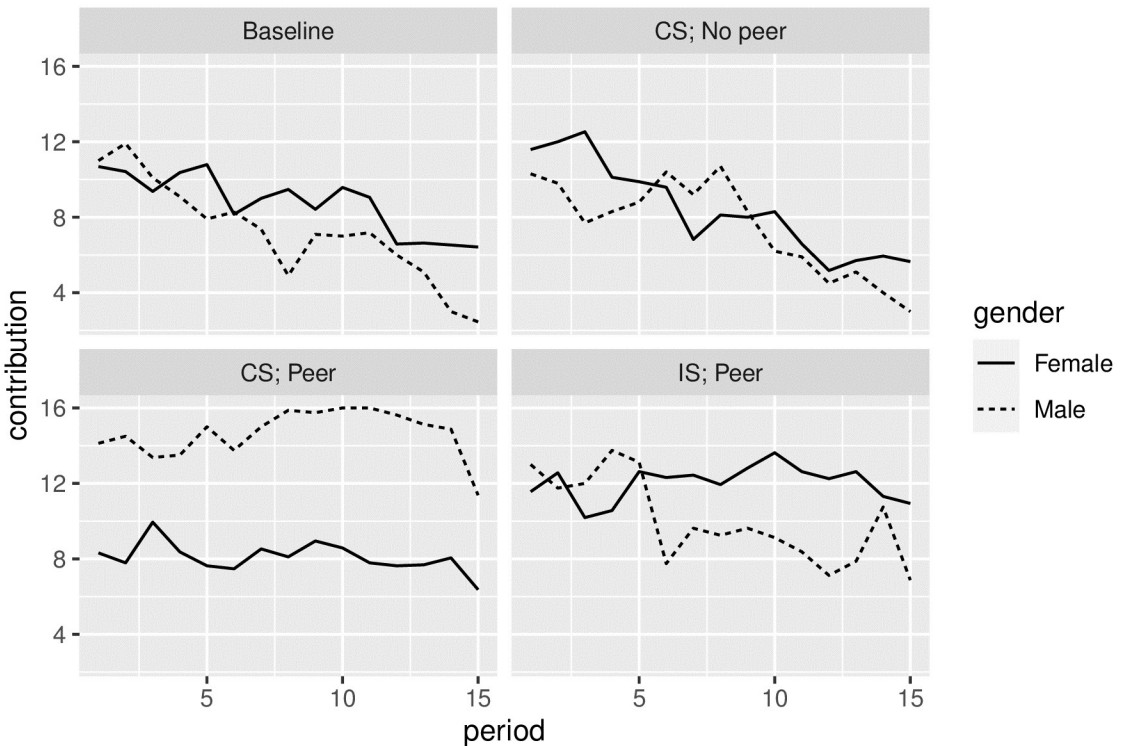

**Fig 5. Contributions across periods by gender.**

43] for using Tobit models for studying PGG data due to the fact that possible contribution levels are bounded from below and above.

Both linear and Tobit panel OLS models use the average contribution of an individual as a dependent variable. While collective sanctions *per se* decreased the average amount contributed, for men the effect is strongly positive. We also included to the models the lagged experience of being sanctioned. It could be expected that the previous experience of sanctions by a central authority would affect participants' behavior in the next round. This reaction is observed in most iterated voluntary contribution experiments, like [44]. The overall effect of an external sanctioning regime can be split into two effects: one from being checked, and one from being punished externally (conditional on one's behavior being checked).

Control and punishment by an external sanctioning authority have different consequences under the collective sanctions (CS) regime as compared to the individual sanctions (IS) regime. Under IS, if the entire group is checked, a person does not bear the external sanctions as long as s/he did not break the rules (in our case, s/he should have contributed more than 10 tokens). Therefore, the external control mechanism can confirm a person's prior beliefs that

**Table 4. Gender differences in contributions.**

| Treatment | % of contributions = T | | % of contributions < T | | Mean contribution | |
|---|---|---|---|---|---|---|
| | Women | Men | Women | Men | Women | Men |
| Baseline | 28 | 32 | 45 | 49 | 8.76 | 7.22 |
| CS; No peer | 30 | 15 | 48 | 57 | 8.40 | 7.48 |
| CS; Peer | 36 | 13 | 43 | 18 | 8.08 | 14.66 |
| IS; Peer | 12 | 62 | 34 | 24 | 12.03 | 10.00 |

**Table 5. DV: Contribution to the group project.** Tobit random-effect baseline and two extended models.

| DV: Contribution | (1) | (2) | (3) |
|---|---|---|---|
| Collective sanctions (CS) | -2.090 | -6.202** | -6.659** |
| | (2.457) | (2.891) | (3.115) |
| Peer sanctions | 3.977 | 4.616** | 4.340* |
| | (2.463) | (2.347) | (2.525) |
| Man | | -5.236 | -6.162* |
| | | (3.440) | (3.692) |
| CS X Man | | 11.24** | 12.15** |
| | | (4.937) | (5.297) |
| Trust | | -6.642*** | -7.405*** |
| | | (2.433) | (2.621) |
| Peer sanctions received$_{t-1}$ | | | 0.131 |
| | | | (0.173) |
| Peer sanctions sent$_{t-1}$ | | | 0.524*** |
| | | | (0.183) |
| CS Applied$_{t-1}$ | | | -1.492 |
| | | | (0.957) |
| Group is checked$_{t-1}$ | | | 2.030*** |
| | | | (0.701) |
| Sigma | 8.208*** | 8.209*** | 7.804*** |
| LL | -3320.37 | -3315.13 | -2985.74 |
| Wald | 3.74 | 15.04*** | 32.55*** |
| Observations | 1,620 | 1,620 | 1,512 |
| Individuals | 108 | 108 | 108 |

Standard errors in parentheses

*** $p < 0.01$,

** $p < 0.05$,

* $p < 0.1$

following the rule is the right decision. However, it may happen that this can provoke the opposite reaction, due to the well-known gambler's fallacy—individuals' believe that an unlikely event becomes less likely in the future when it has just materialized [45].

In model 3 we included lagged variables of the external check at $t - 1$ and external sanctions at $t - 1$. These two lagged variables work in opposite directions: if the group is checked, this increases the investment into a group project in the next period, but if it is checked and punished the contributions drop.

Overall, out of 1,620 individual observations, 1,098 (67.78%) were not checked, while 259 (15.99%) were checked without external sanctions, and 263 (15.23%) were checked and punished externally. Therefore, the groups were checked in 32.22% of the cases, which fits almost perfectly to a predicted 33% level outlined earlier.

Using two binary variables ("External check" and "External punishment"), we constructed a new categorical variable in order to conduct a more fine-grained analysis. Theoretically, the variable can take 2 × 2 values. A group can be (1) "not checked, not punished," (2)"checked, not punished," (3) "checked and punished," and (4) "not checked and punished." However, the last option is not realistically feasible option, leaving us with three, rather than four distinct values. In Fig 6, we used the "No checking" value as our baseline.

**Table 6. DV: Contribution to the group project.** Panel OLS (random-effect) baseline and two extended models.

| DV: Contribution | (1) | (2) | (3) |
|---|---|---|---|
| Collective sanctions (CS) | -0.698 | -2.546* | -2.551*** |
| | (1.172) | (1.402) | (0.965) |
| Peer sanctions | 2.557** | 2.858** | 2.911*** |
| | (1.174) | (1.139) | (0.786) |
| Man | | -2.094 | -2.358** |
| | | (1.678) | (1.154) |
| CS X Man | | 5.032** | 5.270*** |
| | | (2.393) | (1.646) |
| Trust | | -3.133*** | -3.172*** |
| | | (1.180) | (0.812) |
| Peer sanctions received$_{t-1}$ | | | 0.0177 |
| | | | (0.0893) |
| Peer sanctions sent$_{t-1}$ | | | 0.193** |
| | | | (0.0826) |
| CS Applied$_{t-1}$ | | | -1.260*** |
| | | | (0.472) |
| Group is checked$_{t-1}$ | | | 1.185*** |
| | | | (0.350) |
| Sigma | 4.579 | 4.579 | 4.358 |
| $R^2$ | 0.0297 | 0.0890 | 0.119 |
| Wald | 4.967 | 15.95 | 53.82 |
| Observations | 1,620 | 1,620 | 1,512 |
| Individuals | 108 | 108 | 108 |

Standard errors in parentheses

*** $p < 0.01$,

** $p < 0.05$,

* $p < 0.1$

The coefficients of "check, no punishment" and "check, punished" show how deviations from the baseline scenario (no check, no punishment) during the previous period influenced the contributions in the subsequent period. We can see that that subjects reacted differently to external punishment and checks under the two different regimes. Checks of already cooperative subjects (in IS) or groups (in CS) increase cooperation in the next period, even if barely so under IS.

## Fairness evaluation

Men under collective sanctions contributed significantly more than women. This difference can be explained by a gender-based difference in perception of the two regimes. In a post-experimental questionnaire, we asked participants to evaluate the fairness of the specific sanctions rule used in the game. Perception of the regime fairness appears to be a key factor that explains why CS is not as efficient as it should be. If we look at the effect the fairness has on contribution levels (Fig 7) we can see that contributions grow with fairness estimation. Fairness was estimated by participants twice. First, they graded the regime they experienced in the experiment using a four-level Likert scale (from "very unfair" to "totally fair"). Next, we explained the rules of another treatment (collective sanctions to the participants of the

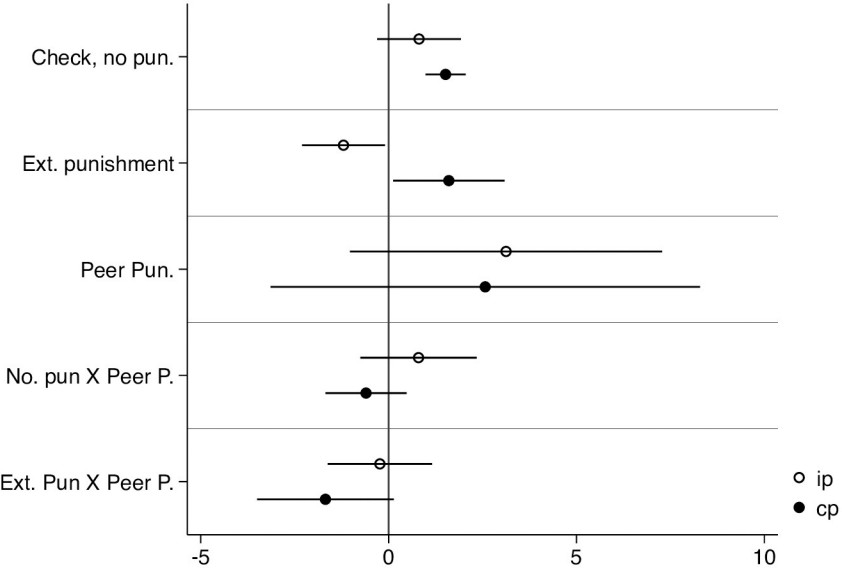

**Fig 6. Panel OLS (random-effect).** DV: Contribution. All IVs are lagged ($t-1$).

individual sanctions regime, and vice versa). Participants then had to grade the fairness of this alternative regime compared to the one they just experienced. This doubled the number of estimations (with all relevant limitations) and usefully put the evaluation of the regime they had experienced into context.

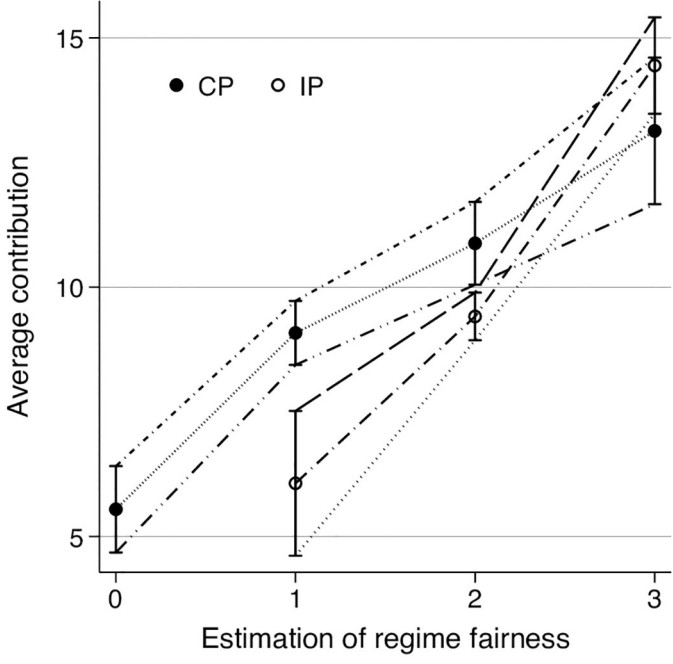

**Fig 7. Contribution levels in different regimes by fairness.**

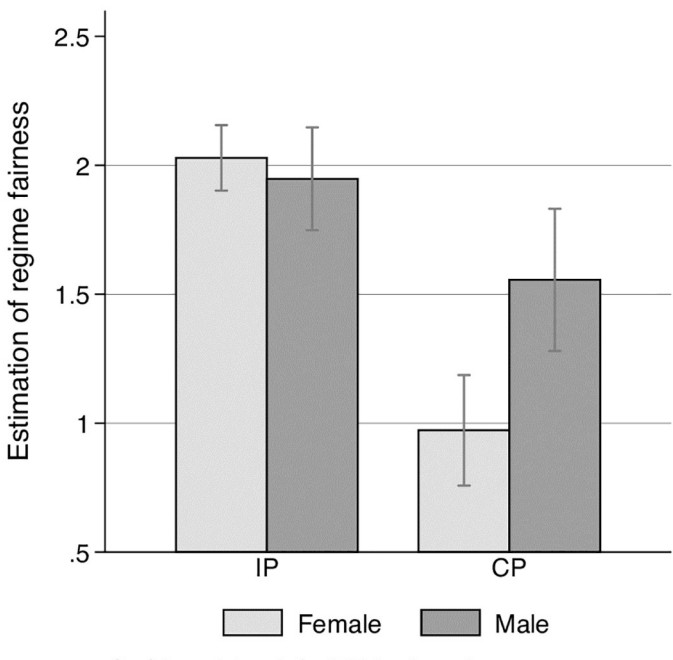

**Fig 8. Fairness estimation by gender in two different regimes.**

While for IS there was almost no difference in the fairness evaluations between men and women, for CS, women found the regime much more unfair; the difference is statistically significant at a 10% level (see Fig 8).

## Discussion

As mentioned in "Theoretical arguments for collective sanctions", excepting one unpublished paper [16], there are very few comparably designed studies. This does not mean there have not been other studies focused on negative incentives imposed upon the entire group. We may treat as a collective sanction any random or 'noisy' sanctioning mechanisms where there is no sure guarantee that the punishment will be applied to its intended target. This interpretation of random sanctions as collective is derived from the concept of expected utility. On this view, if the chance of being punished individually with intensity $F$ is $p$, then we may interpret it as a sanction of intensity $pF$ applied to each group member (assuming risk neutrality of group members). The most comprehensive overview of the legal dimension of collective sanctions agrees: "*so long as groups are sufficiently solidary, group incentives will be the same whether collective sanctions are lumped on one member of the group chosen at random (or by any other criteria besides culpability) or spread evenly among all group members*" [10, p. 367]. From this perspective, [17] appears most similar to our experimental design. There, a randomly chosen member is punished by exclusion from the group and from receiving a share of a public good, if a group on average fails to meet a certain threshold of contribution level. While participants found this approach procedurally unfair, it promoted cooperation significantly. However, in [17], the probability of exclusion grew linearly with the number of violators, which made it rational for participants to cooperate when the expected frequency of violations increased. In this sense, the efficiency of collective sanctions was not tested but was rather implied by design.

Several other experimental studies have explored the efficacy of collective or random sanctions in different dimensions. In [46], the punishment and reward were imposed on an individual with a probability that grew as a function of his deviation from an average in the group. The study found that negative incentives applied in this way are more efficient than positive ones. However, by contrast with our design, this study included a 'noisy' individual sanctioning regime: the probability of being punished or rewarded was based on the individual level of contribution to a public good. Another study [47] used a modified version of Corruption game [48] to investigate whether the threat of collective sanctions imposed upon all public officials who accepted a bribe would prevent individual bribe taking (it did not). Since corrupt deals are 'public bads', and the game assumes asymmetrical roles within a group (Public officials vs. Private citizens), the results of [47] are not directly comparable to our findings.

The other subset of studies of collective sanctions are in the field of social psychology. These studies have a long tradition, beginning with vignette experiments studying children's reaction to collective sanctions [49]. They focus on individual attitudes towards collective sanctions, i.e. the question of their legitimacy and fairness, depending on the context of the situation. Acceptance of and readiness to apply collective sanctions vary with group entitativity (degree of members' similarity) [50], power structure within a group (democratic vs. non-democratic) [1] and intergroup competition [51]. However, unlike the current study, these studies focusing on stated preferences do not capture the effect the threat of collective sanctions may have on individual behavior.

The current study has some evident limitations, however. Since the gender effect was not a primary initial focus of this study, first we need to put its findings into the context of a vast pre-existing literature on gender. There is convincing evidence based on large-scale worldwide surveys that women are more pro-social and less prone to negative reciprocity [52]. If we focus on observed behavioral differences, then no consensus exists regarding gender differences in cooperation in social dilemmas. Two large meta-studies [53, 54] did not find substantial differences between the genders, although men tend to be slightly more cooperative in repeated interactions [53]. Some studies of public good games have found that women contribute more [55], and others have found that men contribute more [56]. Others still found more nuanced effects, such as the observation that women contribute significantly more when the free-riding option is intentionally framed as a harm to the rest of the group [57], or that women start with higher levels of contributions (although the effect fades over time) [58]. While, in this paper, we controlled for income and SAT level, and all the participants were 2nd- and 3rd-year students at Columbia University (thus we have implicitly controlled for educational level and, to a certain extent, age), additional controls are necessary to corroborate our findings.

Second, the framing effect influences individual choices in most of the social dilemmas [59, 60]. To avoid this, we used neutral wording to describe different sanctioning regimes. We thereby relied upon the arguably unrealistic assumption that participants would require no information about what drives the intention of the authority when a specific sanctioning regime is applied, despite the fact that when sanctions are procedurally unfair [61]—or when the intention of the central authority is questionable [62]—this may drastically reduce levels of cooperation.

## Author Contributions

**Data curation:** Philipp Chapkovski.

**Formal analysis:** Philipp Chapkovski.

**Investigation:** Philipp Chapkovski.

**Methodology:** Philipp Chapkovski.

**Software:** Philipp Chapkovski.

**Visualization:** Philipp Chapkovski.

**Writing – original draft:** Philipp Chapkovski.

**Writing – review & editing:** Philipp Chapkovski.

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
