## [Decision Letter · Decision Letter 0]

21 Jan 2021

PONE-D-20-40936

Strike one hundred to educate one: can collective sanctions be efficient?

PLOS ONE

Dear Dr. Chapkovski,

Thank you for submitting your manuscript to PLOS ONE. After careful consideration, we feel that it has merit but does not fully meet PLOS ONE’s publication criteria as it currently stands. Therefore, we invite you to submit a revised version of the manuscript that addresses the points raised during the review process.

The two reviewers have provided constructive and detailed comments. They both agreed that the work is interesting, relevant and would provide a good contribution (to the study of incentives & cooperation). However, there are several aspects of the paper that need improvements, for which the reviewers have provided constructive suggestions. Please carefully consider them in the revision of your manuscript.

We look forward to receiving your revised manuscript.

Kind regards,

The Anh Han, Ph.D.

Academic Editor

PLOS ONE

Journal Requirements:

2. Please note that according to our submission guidelines (http://journals.plos.org/plosone/s/submission-guidelines), outmoded terms and potentially stigmatizing labels should be changed to more current, acceptable terminology. In order to avoid conflation between gender and sex, "female” or "male" should be changed to "woman” or "man" as appropriate, when used as a noun.

Additional Editor Comments:

The two reviewers have provided constructive and detailed comments. They both agreed that the work is interesting, relevant and would provide a good contribution. However, there are several aspects of the paper that need improvements, for which the reviewers have provided constructive suggestions. Please carefully consider them in the revision of your manuscript.

Reviewers' comments:

Reviewer's Responses to Questions

**Comments to the Author**

1. Is the manuscript technically sound, and do the data support the conclusions?

Reviewer #1: Yes

Reviewer #2: Yes

2. Has the statistical analysis been performed appropriately and rigorously? 

Reviewer #1: Yes

Reviewer #2: Yes

3. Have the authors made all data underlying the findings in their manuscript fully available?

Reviewer #1: Yes

Reviewer #2: Yes

4. Is the manuscript presented in an intelligible fashion and written in standard English?

Reviewer #1: Yes

Reviewer #2: No

5. Review Comments to the Author

Reviewer #1: In this manuscript, the author compares the effect of collective sanctions and individual sanctions in promoting contributions by means of behavioral experiment. Importantly, the author answers the question: can collective sanctions for an individual’s antisocial behavior be beneficial for the norm of cooperation? Besides, the author also analyzes the possible reasons for the failure of collective sanctions. Finally, the author analyzes the influence of gender on the results.

There are some remaining issues with the manuscript, requiring some answers.

Major issue:

1) In line 175, the experiment consisted of 15 periods. Here, do the game participants know when the game will end?

2) The explanation of Figure 1 is unclear, including two “Yes” and “N0”.

3) In lines 314 – 323, the descriptions of the results presented in Figure 2 are inaccurate. “Without peer sanctions, cooperation began to decline after the 5th or 6th round to contributions of 5 or 6 tokens out of 20. With peer sanctions, the average contributions remained relatively stable at about half of the endowment (10-12 tokens) until the 15th (and the last) round” From periods 6-11, I can still find that the contribution level is above 6. The description should be more accurate.

4) In the individual sanctions regime, the contributions would be checked. Will this check generate an observation cost?

5) In the process of experiment, the individuals participating in the game have great heterogeneity, such as education level, culture, major, age. Why does the author only explore the influence of gender on the results? Are other variables controlled?

6) In the section “Theoretical arguments for collective sanctions”, the author should clarify the difference between the collective sanctions mentioned in this manuscript and the costly punishment in previous works, such as Emergence of social punishment and cooperation through prior commitments. In AAAI, pp. 2494-2500.

Minor issue:

(1) It is better to use declarative sentence instead of interrogative sentence in the title of the manuscript.

(2) In line 201, perr should change to peer.

(3) In line 201, what does SPGG mean? This abbreviation should be marked where it first appears.

(4) In lines 240-242, the author should describe how the probability of check is set in the case of collective sanctions.

(5) Between 243 and 244, should the pi_i be change to pi_j in the equation of CP?

(6) In line 264, public good game should be corrected as public good games.

(7) In line 272, drop should be corrected as drops.

(8) In line 297 and 303, table should change to Table.

(9) In line 339, gi should be corrected as g_i.

(10) In line 343, represent should be corrected as represents.

Reviewer #2: This paper reports on an experiment testing the effect of collective vs. Individual sanctions on cooperative behaviour in the public goods game. The paper is well motivated and the design of the experiment and the analysis of the results are sound. However, I have several issues with the Discussion and the Literature Review. I think that this paper can be published after a major revision.

I list below the comments that I have taken while reading the manuscript:

- “People tend to cooperate more with their own group members”. This statement needs a reference. I am aware of one paper, making this point in the dictator game (Bilancini et al. 2020), perhaps it could be useful, although dictator game giving is not exactly as public goods cooperation.

- Line 178. “investment” -> “invest”. More generally, please double check the writing. I have noticed several typos.

- Line 181. Were the participants informed that the group was fixed across rounds?

- Formula after line 201. This utility function does not include any peer sanction, so it’s not clear why it is introduced as “Fehr and Schmidt’s public good with peer punishment”. Moreover, the public goods game, in general (as defined by that utility function) was not introduced by Fehr and Schmidt. More generally, I don’t think that formula is useful at all. Every reader of this paper would know the public goods game.

- Similarly, I found the formulas after line 243 pointless. The necessary information are already in the text.

- Line 272-275. The logic around group size is unclear. Note that it is not obvious that larger group size increases larger or the same number of cooperators. Sometimes group size has a positive effect on cooperation (Barcelo & Capraro, 2015; Pereda, Capraro & Sanchez, 2019); other time the effect is curvilinear (Capraro & Barcelo, 2015). This seems relevant and should probably be discussed.

- Table 2. Please eliminate the word “tab” from the description of the table.

- Table 3. I think you want to say “lower bound” and not “lower boundary”. Moreover, you have to tell the confidence interval. In general, lower bound does not make any sense in this context.

- Figure 1. What does “intsanction” mean? Note that figures should be as self-explanatory as possible, to help the reader to understand the key point of the paper without necessarily read all the details.

- Line 312. “participants in CS regime contributed significantly less”. Less than who??

- Line 332. The trimodal distribution was already observed by Capraro, Jordan and Rand (2014). Please discuss the relationship between your paper and theirs. Note that Capraro et al. observed a trimodal distribution in a standard PGG (and argue that participants follow a “give half heuristic”. In any case, the fact that they observe a trimodal distribution in the standard PGG implies that your interpretation that this trimodal distribution is due to the threshold is probably wrong.

- Gender differences. You should discuss the relationship between your result and those of Rand (2017) and Balliet et al, who found gender differences in cooperation in the standard PGG.

- The discussion should be largely rewritten. One of the goals of the discussion section is to compare the current work with previous works. The current discussion has only one reference, so it dramatically fails to make this comparison. In general, I think that this paper largely fails in relating its results with previous work. Another goal of the discussion is to list limitations of the work. The current discussion does not list any limitation. But every experimental work has limitations!

References

Balliet, D., Li, N. P., Macfarlan, S. J., & Van Vugt, M. (2011). Sex differences in cooperation: a meta-analytic review of social dilemmas. Psychological bulletin, 137(6), 881.

Barcelo H, Capraro V (2015) Group size effect on cooperation in one-shot social dilemmas. Scientific Reports 5, 7937.

Bilancini E, Boncinelli L, Capraro V, Celadin T, Di Paolo R (2020) “Do the right thing” for whom? An Experiment on Ingroup Favouritism, Group Assorting and Moral Suasion. Judgment and Decision Making 15, 182-192.

Capraro V, Barcelo H (2015) Group size effect on cooperation in one-shot social dilemmas II. Curvilinear effect. PLoS ONE 10, e0131419.

Pereda M, Capraro V, S ´anchez A (2019) Group size effects and critical mass in public goods games. Scientific Reports 9, 5503.

Capraro V, Jordan JJ, Rand DG (2014) Heuristics guide the implementation of social prefer- ences in one-shot Prisoner’s Dilemma experiments. Scientific Reports 4, 6790.

Rand, D. G. (2017). Social dilemma cooperation (unlike Dictator Game giving) is intuitive for men as well as women. Journal of experimental social psychology, 73, 164-168.

6. PLOS authors have the option to publish the peer review history of their article (what does this mean?). If published, this will include your full peer review and any attached files.

Reviewer #1: No

Reviewer #2: No

---

## [Author Response · Author response to Decision Letter 0]

12 Feb 2021

I am deeply grateful to both reviewers for the time and effort they spent providing extremely helpful and insightful comments on my paper. Below I provide point-by-point responses to each reviewers’ comments and suggestions (starting with major issues). The line numbers refer to the revised manuscript (‘clean copy‘). Based on reviewers suggestions the ‘Discussion’ section was completely re-written, and a ‘Theoretical arguments’ section has undergone significant changes to incorporate reviewers suggestions.

Both reviewers noticed a number of typos and errors in the English language. Therefore, in addition to correcting specific errors mentioned by reviewers, I also additionally checked the entire text again for the flaws and typos in English language. 

In accordance with the submission guidelines of PLOS ONE I also changed "female" or "male" to "woman" or "man" where it was used as a noun everywhere in the text.

Reviewer #1. Major issues:

=====

1) In line 175, the experiment consisted of 15 periods. Here, do the game participants know when the game will end?

Thanks! The question how much did participant know was also raised by a Reviewer #2. Yes, that’s why we observe a end-game effect. Based on both your and Reviewer #2 comments I described what was known to the participants before the game in lines 197-205. 

=====

2) The explanation of Figure 1 is unclear, including two “Yes” and “N0”.

Thank you for point this. Figure 1 is re-done to include all the relevant information.

=====

3) In lines 314 – 323, the descriptions of the results presented in Figure 2 are inaccurate. “Without peer sanctions, cooperation began to decline after the 5th or 6th round to contributions of 5 or 6 tokens out of 20. With peer sanctions, the average contributions remained relatively stable at about half of the endowment (10-12 tokens) until the 15th (and the last) round” From periods 6-11, I can still find that the contribution level is above 6. The description should be more accurate.

Thanks again for noticing this. I slightly re-did Figure 2 (without changing its content) to make the difference between treatments more visible. I also corrected the text based on your comments (lines 352-357).

=====

4) In the individual sanctions regime, the contributions would be checked. Will this check generate an observation cost?

I entirely agree that I mentioned the ‘informational dimension’ of collective sanctions but later on do not use this factor in the experimental design, so the check did not generate an observation cost. That was done mostly for the sake of simplicity and based on your comment I provide the explanation why I did it this way in lines 249-255, ‘Experimental design’ section. 

=====

5) In the process of experiment, the individuals participating in the game have great heterogeneity, such as education level, culture, major, age. Why does the author only explore the influence of gender on the results? Are other variables controlled?

That is a very important point, which had not been covered in the original version of the text. This ‘gender’ effect was controlled for income and age, but since the difference between genders was not the initial focus of this study, of course more rigorous controls are needed. I mention this when I describe the limitations of this study in the ‘Discussion’ section (lines 498-515).

=====

6) In the section “Theoretical arguments for collective sanctions”, the author should clarify the difference between the collective sanctions mentioned in this manuscript and the costly punishment in previous works, such as Emergence of social punishment and cooperation through prior commitments. In AAAI, pp. 2494-2500.

Yes, I totally agree. I added the paragraph with a current state of the art on sanctions in social dilemmas to a ‘Theoretical arguments’ section (lines 60-73), including among others the referred paper.

Reviewer #1. Minor issues

(1) It is better to use declarative sentence instead of interrogative sentence in the title of the manuscript.

That is an excellent suggestion. The title was changed from “Strike one hundred to educate one: can collective sanctions be efficient?” to a new one “Strike one hundred to educate one: measuring the efficacy of collective sanctions experimentally”

=====

(3) In line 201, what does SPGG mean? This abbreviation should be marked where it first appears.

I deleted this abbreviation since it was not used nowhere in the text below. 

=====

(4) In lines 240-242, the author should describe how the probability of check is set in the case of collective sanctions.

The detailed explanation was included into lines 261-267.

=====

(5) Between 243 and 244, should the pi_i be change to pi_j in the equation of CP?

Based on Reviewer #2 suggestions I completely eliminated the formulae mentioned above and incorporated them in a more compact way into Equation (1) - line 285, correcting this typo as well. 

=====

(2) In line 201, perr should change to peer.

(6) In line 264, public good game should be corrected as public good games.

(7) In line 272, drop should be corrected as drops.

(8) In line 297 and 303, table should change to Table.

(9) In line 339, gi should be corrected as g_i.

(10) In line 343, represent should be corrected as represents.

These typos (along with others) was corrected during the additional proofreading

Reviewer #2. 

- “People tend to cooperate more with their own group members”. This statement needs a reference. I am aware of one paper, making this point in the dictator game (Bilancini et al. 2020), perhaps it could be useful, although dictator game giving is not exactly as public goods cooperation.

Thank you very much for this comment! In a mostly re-written ‘Theoretical arguments’ section I expanded (lines 123-133) the part where I discuss this ingroup bias in cooperation, including among other works the paper by Bilancini et al. mentioned above. 

=======

- Line 178. “investment” -> “invest”. More generally, please double check the writing. I have noticed several typos.

Thank you. I have to apologize for numerous typos and errors in English which were in the original version. I proof-read once again the entire text cleaning the typos.

=======

- Line 181. Were the participants informed that the group was fixed across rounds?

Thank you, this is very important question, and it was raised by a Reviewer #1 as well. Yes, they were informed that the group membership is fixed and stays the same across all 15 rounds. Based on both your and Reviewer #1 comments I described what was known to the participants before the game in lines 197-205. 

========

- Formula after line 201. This utility function does not include any peer sanction, so it’s not clear why it is introduced as “Fehr and Schmidt’s public good with peer punishment”. Moreover, the public goods game, in general (as defined by that utility function) was not introduced by Fehr and Schmidt. More generally, I don’t think that formula is useful at all. Every reader of this paper would know the public goods game.

- Similarly, I found the formulas after line 243 pointless. The necessary information are already in the text.

Based on your comments I deleted formulae mentioned above. Still I decided to include one ‘compact’ formula that sums up the final payoff calculation for all treatments (lines 275-285 and Equation 1 at line 285). Although I agree that this information is redundant, I think that some readers may find the ‘formal’ description helpful and more ‘readable’ than just a descriptive text. If you or other reviewers believe that it is unnecessary, I am happy to delete these lines. 

========

- Line 272-275. The logic around group size is unclear. Note that it is not obvious that larger group size increases larger or the same number of cooperators. Sometimes group size has a positive effect on cooperation (Barcelo & Capraro, 2015; Pereda, Capraro & Sanchez, 2019); other time the effect is curvilinear (Capraro & Barcelo, 2015). This seems relevant and should probably be discussed.

Thank you! I decided to elaborate on this in the ‘Hypotheses’ section (lines 146-158) where I provide a compact review of the literature considering group size effect on cooperation, including the papers suggested above. 

========

- Table 2. Please eliminate the word “tab” from the description of the table.

This error is corrected in a revised version, thank you for noticing this!

========

- Table 3. I think you want to say “lower bound” and not “lower boundary”. Moreover, you have to tell the confidence interval. In general, lower bound does not make any sense in this context.

Thank you for noticing that, that was corrected in a revised version: I replaced this info with 95% confidence intervals. 

========

- Figure 1. What does “intsanction” mean? Note that figures should be as 

self-explanatory as possible, to help the reader to understand the key point of the paper without necessarily read all the details.

Yes, I entirely agree, this was also noted by Reviewer #1. Based on your and the other reviewer’s suggestions, I re-formatted Figure 1 entirely to include all the relevant information. 

========

- Line 312. “participants in CS regime contributed significantly less”. Less than who??

This unclarity is now corrected (line 350)

========

- Line 332. The trimodal distribution was already observed by Capraro, Jordan and Rand (2014). Please discuss the relationship between your paper and theirs. Note that Capraro et al. observed a trimodal distribution in a standard PGG (and argue that participants follow a “give half heuristic”. In any case, the fact that they observe a trimodal distribution in the standard PGG implies that your interpretation that this trimodal distribution is due to the threshold is probably wrong.

Thank you for this comment - that is my fault that I somehow overlooked this paper (Capraro, Jordan and Rand, 2014) which of course extremely important if I’d like to draw any conclusions from an observed trimodal distribution. I included the reference to this paper, and described the limitations of my analysis based on this in lines 369-376. 

=========

- Gender differences. You should discuss the relationship between your result and those of Rand (2017) and Balliet et al, who found gender differences in cooperation in the standard PGG.

In a new version of ‘Discussion’ section, lines 499-515 I provide a quick overview of the current findings in gender differences in social dilemmas, including standard PGGs, and I also include among other papers a paper by Rand and a meta-review of Balliet et al.

=========

- The discussion should be largely rewritten. One of the goals of the discussion section is to compare the current work with previous works. The current discussion has only one reference, so it dramatically fails to make this comparison. In general, I think that this paper largely fails in relating its results with previous work. Another goal of the discussion is to list limitations of the work. The current discussion does not list any limitation. But every experimental work has limitations!

Thank for this comment! Following your suggestion, I entirely re-wrote the entire ‘Discussion’ section. Now it roughly consists of two parts. In the beginning of this section I refer to other similar studies in this field comparing my design with these studies. In the second part I briefly delineate the most important limitations.

---

## [Decision Letter · Decision Letter 1]

25 Feb 2021

PONE-D-20-40936R1

Strike one hundred to educate one: measuring the efficacy of collective sanctions experimentally

PLOS ONE

Dear Dr. Chapkovski,

Thank you for submitting your manuscript to PLOS ONE. After careful consideration, we feel that it has merit but does not fully meet PLOS ONE’s publication criteria as it currently stands. Therefore, we invite you to submit a revised version of the manuscript that addresses the points raised during the review process.

ACADEMIC EDITOR: Both reviewers are happy with the changes made by the authors, and recommended publication subject to some minor revisions. Please take them into account when preparing the revised version of your paper.

We look forward to receiving your revised manuscript.

Kind regards,

The Anh Han, Ph.D.

Academic Editor

PLOS ONE

Journal Requirements:

Additional Editor Comments (if provided):

Both reviewers are happy with the changes made the authors, and recommend publication subject to some minor revisions. Please take them into account when preparing the revised version of your paper.

Reviewers' comments:

Reviewer's Responses to Questions

**Comments to the Author**

1. If the authors have adequately addressed your comments raised in a previous round of review and you feel that this manuscript is now acceptable for publication, you may indicate that here to bypass the “Comments to the Author” section, enter your conflict of interest statement in the “Confidential to Editor” section, and submit your "Accept" recommendation.

Reviewer #1: (No Response)

Reviewer #2: All comments have been addressed

2. Is the manuscript technically sound, and do the data support the conclusions?

Reviewer #1: Yes

Reviewer #2: Yes

3. Has the statistical analysis been performed appropriately and rigorously? 

Reviewer #1: Yes

Reviewer #2: Yes

4. Have the authors made all data underlying the findings in their manuscript fully available?

Reviewer #1: Yes

Reviewer #2: Yes

5. Is the manuscript presented in an intelligible fashion and written in standard English?

Reviewer #1: No

Reviewer #2: Yes

6. Review Comments to the Author

Reviewer #1: The authors have addressed most of my comments, but a few issues with the paper's clarity still remain for me and should be seen to before publication.

For one, there are still some grammatical and tense problems in the revised version. For example, in page 1, line 3: “define”; in page 4, line 149-152: the format of the front quotation marks needs to be adjusted; in page 5, line 211: “N-person Prisoner’s dilemma” “N” should be italicized;

In addition, an important literature on collective punishment (implicated punishment) has not been cited. Evolution of public cooperation in a monitored society with implicated punishment and within-group enforcement. Scientific Reports, 5 (1), 1-12.

Standing initial of the journal name in the reference should be capitalized, such as, 34-35, 40, 44-45, 49, 52.

Reviewer #2: The authors have addressed all my comments.

7. PLOS authors have the option to publish the peer review history of their article (what does this mean?). If published, this will include your full peer review and any attached files.

Reviewer #1: No

Reviewer #2: No

---

## [Author Response · Author response to Decision Letter 1]

25 Feb 2021

Thank you for giving me the opportunity to submit a revised draft (second revision) of the manuscript . 

The comments of a reviewer were very helpful. I fixed grammatical and tense problems mentioned by a Reviewer (and I have noticed and corrected a few others) as well as some inconsistencies in the bibliography. I am particularly grateful to a reviewer for referring me to a paper by Chen, Sasaki and Perc (2015) which is of fundamental importance for the collective sanctions problem, and still I somehow missed it. Now I am referring to their results, both in `Method` and `Introduction` sections. 

Reviewer #1. Minor issues:

=====

For one, there are still some grammatical and tense problems in the revised version. For example, in page 1, line 3: “define”; in page 4, line 149-152: the format of the front quotation marks needs to be adjusted; in page 5, line 211: “N-person Prisoner’s dilemma” “N” should be italicized;

Thank you. I fixed these forementioned and some other grammatical and syntactical errors.

=====

In addition, an important literature on collective punishment (implicated punishment) has not been cited. Evolution of public cooperation in a monitored society with implicated punishment and within-group enforcement. Scientific Reports, 5 (1), 1-12.

That is an extremely valuable reference. I referred to it several times in a revised version of the paper. Thank you!

=====

Standing initial of the journal name in the reference should be capitalized, such as, 34-35, 40, 44-45, 49, 52.

I checked for inconsistencies in bibliography fixing the errors mentioned by you.

---

## [Editor Report · Decision Letter 2]

2 Mar 2021

Strike one hundred to educate one: measuring the efficacy of collective sanctions experimentally

PONE-D-20-40936R2

Dear Dr. Chapkovski,

We’re pleased to inform you that your manuscript has been judged scientifically suitable for publication and will be formally accepted for publication once it meets all outstanding technical requirements.

Kind regards,

The Anh Han, Ph.D.

Academic Editor

PLOS ONE
---

## [Editor Report · Acceptance letter]

16 Mar 2021

PONE-D-20-40936R2 

Strike one hundred to educate one: measuring the efficacy of collective sanctions experimentally 

Dear Dr. Chapkovski:

I'm pleased to inform you that your manuscript has been deemed suitable for publication in PLOS ONE. Congratulations! Your manuscript is now with our production department. 

Kind regards, 

on behalf of

Dr. The Anh Han 

Academic Editor

PLOS ONE